# Parameter Extraction of Solar Photovoltaic Model Based on Nutcracker Optimization Algorithm

**Zhenjiang Duan [1], Hui Yu [2], Qi Zhang [2] and Li Tian [1,*]**

1 Department of Microelectronic Science and Technology, Harbin Institute of Technology, Harbin 150001, China; 22s021047@stu.hit.edu.cn
2 Tianjin Institute of Power Sources, Tianjin 300381, China; robert_yuhui@aliyun.com (H.Y.); fred130qi@sina.com (Q.Z.)
* Correspondence: tianli@hit.edu.cn; Tel.: +86-451-86413451

**Abstract:** In order to improve the accuracy and reliability of the photovoltaic (PV) model, this paper explores a novel nature-inspired metaheuristic algorithm, i.e., the nutcracker optimizer algorithm (NOA), for the parameter extraction of a PV model, such as a single diode model (SDM), double diode model (DDM), and triple diode model (TDM) of PV components. The Aleo Solar S79Y300 monocrystalline silicon solar panel was tested at 1000 W/m$^2$ solar irradiance and 25 °C temperature, and the results of the proposed NOA algorithm were compared with three popular algorithms, i.e., particle swarm optimization (PSO), firework algorithm (FWA), and whale optimization algorithm (WOA), in terms of algorithm accuracy and running time, and non-parametric tests were performed. The results show that the NOA can improve the efficiency of PV parameter extraction, and its performance is the best among the tested algorithms. It has the best root mean square error (RMSE) values in the SDM, being $7.92587 \times 10^{-5}$ and $6.02460 \times 10^{-5}$ in the DDM and $6.23617 \times 10^{-5}$ in the TDM, and the shortest average execution time according to the overall ranking, making it well suited for extracting PV model parameters.

**Keywords:** nutcracker optimization algorithm (NOA); photovoltaic (PV) model; parameter extraction; metaheuristic algorithms





## 1. Introduction

In order to tackle the well-known challenges of energy transition, a swift shift from fossil fuels to a wider renewable energy period is required; solar energy is the most abundant and easily accessible renewable energy resource. Annually, the total amount of solar radiation received by the Earth is 7500 times the global energy consumption (about 450 EJ), thus the utilization of solar energy is playing an increasingly important role in the global energy transformation [1]. Photovoltaic researchers have been striving to achieve reliable and accurate modeling of PV components for a better understanding of their operating principles, performance, and characteristics, as well as predicting their performance and lifespan under different environmental conditions. The single diode model (SDM) is the foundation of photovoltaic component modeling, while the double diode model (DDM) considers the recombination loss in the depletion region of the PN junction based on the SDM, and the triple diode model (TDM) is considered to be a more accurate model than the SDM and DDM [2]. It introduces a third diode to characterize solar cells under reverse current conditions [2,3], and it contains nine parameters: $I_{ph}$, $I_{SD1}$, $I_{SD2}$, $I_{SD3}$, $n_1$, $n_2$, $n_3$, $Rs$, and $Rp$. Due to the nonlinearity and complexity of the model, most mathematical and numerical analysis methods are simplified by certain approximation. Metaheuristic algorithms, well suited for solving complex optimization problems with large search spaces and non-linear objective functions, have been extensively studied in the field. In previous research, a diverse range of algorithms and their variants, including genetic algorithm (GA) [4], particle swarm optimization (PSO) [5], differential evolution (DE) [6],

reinforcement learning neural network algorithm (RLNNA) [7], simplified bird mating optimization (SBMO) [8], firework algorithm (FWA) [9], artificial bee colony (ABC) [10], moth flame optimization (MFO) [11], harmony search algorithm (Hs) [12], mutative-scale parallel chaos optimization algorithm (MPCOA) [13], repairing self-adaptive differential evolution (Rcr-IJADE) [14], bee pollinator flower pollination algorithm (BPFPA) [15], free search differential evolution algorithm (FSDE) [16], teaching-learning based optimization (TLBO) [17], Generalized opposition-based learning teaching-learning based optimization (GOTLBO) [18], and evaporation rate-based water cycle algorithm (ER-WCA) [19], have been utilized for the parameter extraction of solar PV models. Recent trends in research also indicate an increasingly frequent pivot towards the exploration and utilization of novel metaheuristic algorithms. For example, Faiz Ali et al. proposed the use of the atomic orbital search metaheuristic algorithm (AOS) for solar cell parameter identification [20], which simulates the mixing and recombination of atomic orbitals to search for the optimal solution. They also discussed more than 30 different RMSE calculation methods to achieve a novel and accurate RMSE calculation. Youssef Kharchouf et al. proposed an improved DE algorithm for parameter extraction of photovoltaic cells in 2022 [21]. He used a numerical method for the Lambert W function analytical equation to improve the local convergence characteristics of DE, which reduced the computation time to half of its original time. Xiangchen et al. proposed a two-stage identification method [22]. First, they preprocessed the experimental I–V curve by removing outliers, curve fitting, and the sparsification of the I–V data. Following this, they used maximum power matching (MPM) to roughly identify the parameters from the preprocessed data. Finally, they used the results as the initial values for the improved flow direction algorithm (IFDA) iteration to achieve the accurate identification of photovoltaic model parameters. Amr A. Abd El-Mageed et al. proposed an improved queuing search optimization (QSO) algorithm based on DE technology and boundary constraint correction process [23], which identified the parameters of the SDM and DDM models. They used DE to increase the population diversity of IQSODE, thereby improving the exploration of the algorithm in the search space and solving the problem of the QSO algorithm being trapped in local optima. Junfeng Zhou et al. proposed a dynamic reverse learning strategy for identifying photovoltaic cell model parameters using an adaptive differential evolution algorithm [24]. The strategy expanded the search range of particles in the global optimal solution search phase, thereby increasing the probability of particles approaching the optimal solution area.

Previous research has mainly focused on one or two of the three models, yet this study presents a novel approach by employing a nature-inspired metaheuristic method known as the nutcracker optimization algorithm (NOA) [25] for the parameter extraction of single, double, and triple diode models. Aleo Solar S79Y300 monocrystalline silicon solar panels were tested under 1000 W/m$^2$ solar radiation and 25 °C temperature. These conditions are standard testing conditions and commonly encountered in many geographical regions globally where substantial investment in solar energy is being made, thereby ensuring our findings have practical and relevant applications. We evaluated the performance of NOA for PV parameter extraction by observing the RMSE error and running time. Subsequent non-parametric tests were performed to analyze the data further. Finally, a comparative analysis was conducted between the proposed NOA and three popular algorithms, PSO [26], FWA [27], and WOA [28], summarizing the results for comprehensive understanding.

The outstanding contributions of the present article are the following:

- Demonstrate the suitability and effectiveness of NOA in the field of the parameter extraction of the PV model.
- Employ modern techniques to accurately identify the optimum parameters in the PV model.
- Conduct a comparative study on four MAs to enhance the precision of a specific diode model optimization.
- Analyze possible future trends in the area based on the test results.

## 2. Solar Cell Model

### 2.1. Photovoltaic (PV) Cell Models

The models of photovoltaic (PV) cells mainly include a series of mathematical models, such as the single diode model (SDM), double diode model (DDM), triple diode model (TDM), diode models with reverse bias characteristics [29], and the improved double diode model (MDDSCM) [30,31], and so on. In this study, we adopted the most commonly used models: SDM, DDM, and TDM. Figure 1 and Table 1 show the circuit structure and equations (including the circuit equation of the PV component) of the SDM, DDM, and TDM. In the equations, $I_L$ represents the output current, $I_{ph}$ represents the photocurrent of the PV component, $I_{SD1}$, $I_{SD2}$, and $I_{SD3}$ represent the reverse saturation current, $Ns$ represents the number of cells connected in series, $Np$ represents the number of cells connected in parallel, $V_T$ (=$n*k*T/q$) is the thermal voltage of the PV component, $q$ is the electron charge ($1.60217646 \times 10^{-19}$ C), $k$ is the Boltzmann constant ($1.3806503 \times 10^{-23}$ J/k), $T$ is the temperature of the PN junction (in K), $n$ is the ideality factor of the diode, $Rp$ is the parallel resistance, $Rs$ is the series resistance, and $R_L$ is the external load resistance.

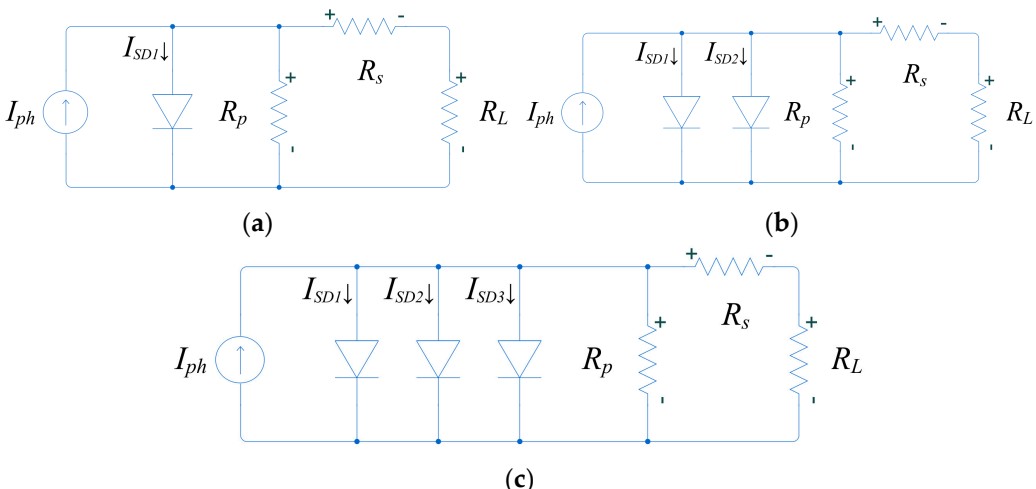

**Figure 1.** Equivalent circuit of (**a**) SDM, (**b**) DDM, (**c**) TDM.

**Table 1.** Circuit equations of SDM, DDM, TDM.

| Model | Circuit Equations |
|---|---|
| SDM | $I_L = I_{ph} - I_{SD1}\left[\exp\left(\dfrac{q(V_L + I_L R_s)}{n_1 kT}\right) - 1\right] - \left[\dfrac{V_L + I_L R_s}{R_{sh}}\right]$ |
| DDM | $I_L = I_{ph} - I_{SD1}\left[\exp\left(\dfrac{q(V_L + I_L R_s)}{n_1 kT}\right) - 1\right] - I_{SD2}\left[\exp\left(\dfrac{q(V_L + I_L R_s)}{n_2 kT}\right) - 1\right] - \left[\dfrac{V_L + I_L R_s}{R_{sh}}\right]$ |
| TDM | $I_L = I_{ph} - I_{SD1}\left[\exp\left(\dfrac{q(V_L + I_L R_s)}{n_1 kT}\right) - 1\right] - I_{SD2}\left[\exp\left(\dfrac{q(V_L + I_L R_s)}{n_2 kT}\right) - 1\right] - I_{SD3}\left[\exp\left(\dfrac{q(V_L + I_L R_s)}{n_3 kT}\right) - 1\right] - \left[\dfrac{V_L + I_L R_s}{R_{sh}}\right]$ |
| PV | $I_L = I_{ph}N_p - I_{SD}N_p\left[\exp\left(\dfrac{q\left(V_L + \dfrac{N_s I_L R_s}{N_p}\right)}{nkN_S T}\right) - 1\right] - \left[\left(V_L + \dfrac{N_s I_L R_s}{N_p}\right) \Big/ \dfrac{R_{sh}N_s}{N_p}\right]$ |

### 2.2. Optimization of Photovoltaic (PV) Cell Model Parameters

The optimization of photovoltaic (PV) cell model parameters using metaheuristic algorithms can be understood as a process of optimizing a single objective function. The objective function is typically defined as an error function that measures the discrepancy between the model's predictions and experimental data. The optimization algorithm iteratively searches for the optimal set of parameters that minimize the error function. The resulting solution is a set of optimized parameter values, denoted as $X_S$, $X_D$, and $X_T$.

Finally, the optimized parameter values are validated by comparing the simulated model output to the result of experimental measurements.

In this paper, the root mean square error (RMSE) is applied as the fitness function for algorithm optimization. The error function is represented by the RMSE shown in Table 2 which quantifies the accuracy and goodness of fit of the proposed current–voltage characteristic model. By experimentally measuring a set of actual current values ($I_{mea}$) of solar cell modules under equidistant voltages, feasible solutions are obtained through MA optimization and then incorporated into the optimization functions of SDM, DDM, and TDM in Table 3. Consequently, the $I_{cal}$ value is calculated, and the fitness function value is derived by incorporating it into the RMSE formula.

**Table 2.** Formula for the error function.

| Evaluation Index | Formula |
|:---:|:---:|
| RMSE | $\text{RMSE} = \sqrt{\frac{1}{N}\sum_{i=1}^{N}\left(I_{mea} - I_{cal}\right)^2}$ [1] |

[1] $I_{mea}$ represents the actual current value obtained from the experiment, and $I_{cal}$ represents the calculated current value after model optimization.

**Table 3.** Parameter optimization functions for SDM, DDM, and TDM.

| Model | Parameter Optimization Functions |
|:---:|:---:|
| SDM | $f_S(V_t, I_t, X_s) = I_L - x_3 + x_4\left[\left(\frac{q(V_t + x_1 \cdot I_t)}{x_5 \cdot k \cdot T}\right) - 1\right] + \frac{V_t + x_1 \cdot I_t}{x_2}$ |
| DDM | $f_D(V_t, I_t, X_D) = I_L - x_3 + x_4\left[\left(\frac{q(V_t + x_1 \cdot I_t)}{x_6 \cdot k \cdot T}\right) - 1\right] + x_5\left[\left(\frac{q(V_t + x_1 \cdot I_t)}{x_7 \cdot k \cdot T}\right) - 1\right] + \frac{V_t + x_1 \cdot I_t}{x_2}$ |
| TDM | $f_T(V_t, I_t, X_T) = I_L - x_3 + x_4\left[\left(\frac{q(V_t + x_1 \cdot I_t)}{x_6 \cdot k \cdot T}\right) - 1\right] + x_5\left[\left(\frac{q(V_t + x_1 \cdot I_t)}{x_7 \cdot k \cdot T}\right) - 1\right] + x_8\left[\left(\frac{q(V_t + x_1 \cdot I_t)}{x_9 \cdot k \cdot T}\right) - 1\right] + \frac{V_t + x_1 \cdot I_t}{x_2}$ |

$X_s = [R_s, R_{sh}, I_{ph}, I_{SD1}, n_1]$, $X_D = [R_s, R_{sh}, I_{ph}, I_{SD1}, I_{SD2}, n_1, n_2]$, $X_T = [R_s, R_{sh}, I_{ph}, I_{SD1}, I_{SD2}, I_{SD3}, n_1, n_2, n_3]$.

## 3. NOA

### 3.1. Overview of NOA

NOA is a novel nature-inspired metaheuristic algorithm (MA) inspired by the Clark's nutcracker [25]. The nutcracker collects pine nuts (food) in the summer and autumn and stores them in specific locations, then retrieves the storage location and searches for food in the spring and winter. Inspired by this idea of food access, we proposed two strategies: (i) a foraging and caching strategy and (ii) a caching search and recovery strategy. After population initialization, random exploration and exploitation optimization are carried out, and RMSE is the fitness function. The nomenclature used in the NOA algorithm is shown in Table 4.

**Table 4.** The nomenclature used in the NOA algorithm.

| Parameters | Description |
|:---|:---|
| $\overrightarrow{X}_i^t$ | The current position/current cache of nutcrackers in iteration t. |
| $X_{i,j}^t$ | The $j$th position of the $i$th nutcracker in the current generation. |
| $U_j, L_j$ | Vectors, including the upper and lower bound of the $j$th dimension in the optimization problem. |
| $\gamma, \lambda$ | A number generated according to the Lévy flight. |
| $\overrightarrow{X}_{best}^t$ | The $j$th dimension of the best solution obtained. |
| $\tau, r, \varphi$ | Random real numbers in the range of [0, 1]. |
| $X_{m,j}^t$ | The mean of the jth dimensions of all solutions of the current population in the iteration t. |
| $\overrightarrow{X}^{t+1(new)}$ | A new position in the storage area of the nutcrackers in current iteration t. |
| $l$ | A factor that linearly decreased from 1 to 0. |
| $P_{a_1}$ | A probability value that is linearly decreased from one to zero. |

**Table 4.** *Cont.*

| Parameters | Description |
| --- | --- |
| $\overrightarrow{RP}_{i,1}^{t}$, $\overrightarrow{RP}_{i,2}^{t}$ | RPs (objects) of the cache position $\overrightarrow{X}_{i}^{t}$ of the ith nutcracker in the current generation t. |
| $\theta$ | The angle-of-view of the nutcracker, chosen at random from $[0, \pi]$. |
| $t$, $T_{max}$ | The current and maximum generations. |
| $P_{a_2}$ | A probability value that is equal to 0.2. |
| $\overrightarrow{RM}$ | A random vector in the interval $[0, 1]$. |

### 3.2. Foraging and Storage Strategy

The nutcracker searches for good seeds during summer and autumn (first exploration phase), and stores them in a storage area (first exploitation strategy). Due to harsh weather conditions such as heavy snow and wind, the storage area is often far from the search area and has less vegetation, making it easier for the nutcracker to retrieve the seeds during winter and spring.

$$\overrightarrow{X}_i^{t+1} = \begin{cases} X_{i,j}^t & if\ \tau_1 < \tau_2 \\ \begin{cases} X_{m,j}^t + \gamma \cdot \left(X_{A,j}^t - X_{B,j}^t\right) + \mu \cdot (r^2 \cdot U_j - L_j) & if\ t \le T_{max}/2.0 \\ X_{C,j}^t + \mu \cdot \left(X_{A,j}^t - X_{B,j}^t\right) + \mu \cdot (r_1 < \delta) \cdot (r^2 \cdot U_j - L_j), & Otherwise \end{cases}, Otherwise \end{cases} \tag{1}$$

$$\mu = \begin{cases} \tau_3 & if\ r_1 < r_2 \\ \tau_4 & if\ r_2 < r_3 \\ \tau_5 & if\ r_1 < r_3 \end{cases} \tag{2}$$

$$\overrightarrow{X}^{t+1(new)} = \begin{cases} \overrightarrow{X}_i^t + \mu \cdot (\overrightarrow{X}_{best}^t - \overrightarrow{X}_i^t) \cdot |\lambda| + r_1 \cdot (\overrightarrow{X}_A^t - \overrightarrow{X}_B^t) & if\ \tau_1 < \tau_2 \\ \overrightarrow{X}_{best}^t + \mu \cdot (\overrightarrow{X}_A^t - \overrightarrow{X}_B^t) & if\ \tau_1 < \tau_3 \\ \overrightarrow{X}_{best}^t \cdot l & Otherwise \end{cases} \tag{3}$$

$$\overrightarrow{X}_i^{t+1} = \begin{cases} Equation\ (1), & if\ \varphi > P_{a_1} \\ Equation\ (3), & Otherwise \end{cases} \tag{4}$$

Formulas (1)–(4) provide a mathematical model for this strategy. Formulas (1) and (2) describe the process of the nutcracker searching for good seeds. The first state of $\overrightarrow{X}_i^{t+1}$ represents maintaining the current optimal solution, and the second state represents exploring the solution space based on the average value $X_{m,j}^t$ of all solutions in the current population, and any solution $X_{A,j}^t$ and $X_{B,j}^t$. Formula (3) simulates the process of the nutcracker storing seeds, which is the first development stage. The algorithm development process is controlled by the contemporary optimal solution $\overrightarrow{X}_{best}^t$ and a variable $l$ that decreases linearly from 1 to 0. Formula (4) represents the balance between exploration and development of the algorithm according to the probability value $P_{a_1}$, which decreases linearly from 1 to 0.

### 3.3. Cache Search and Recovery Strategy

During the spring and winter seasons, the nutcracker retrieves stored food based on spatial memory, using reference points to locate the stored food as their main source of

nutrition. However, approximately 20% of nutcrackers are unable to retrieve stored food, as reported by [32], leading them to search for new food sources.

$$
RPs = \begin{bmatrix} \vec{RP}_{1,1}^{\,t} & \vec{RP}_{1,2}^{\,t} \\ \vdots & \vdots \\ \vec{RP}_{i,1}^{\,t} & \vec{RP}_{i,2}^{\,t} \\ \vdots & \vdots \\ \vec{RP}_{N,1} & \vec{RP}_{N,1}^{\,t} \\ \vdots & \vdots \end{bmatrix} \tag{5}
$$

$$
\vec{RP}_{i,1}^{\,t} = \begin{cases} \vec{X}_i^{\,t} + \alpha \cdot \cos(\theta) \cdot \left( \left( \vec{X}_A^{\,t} - \vec{X}_B^{\,t} \right) \right) + \alpha \cdot \vec{RP}, & if\ \theta = \pi/2 \\ \vec{X}_i^{\,t} + \alpha \cdot \cos(\theta) \cdot \left( \left( \vec{X}_A^{\,t} - \vec{X}_B^{\,t} \right) \right), & Otherwise \end{cases} \tag{6}
$$

$$
\vec{RP}_{i,2}^{\,t} = \begin{cases} \vec{X}_1^{\,t} + \left( \alpha \cdot \cos(\theta) \cdot \left( \left( \vec{U} - \vec{U} \right) \cdot \tau_3 + \vec{U} \right) + \alpha \cdot \vec{RP} \right) \cdot \vec{U}_2, & if\ \theta = \pi/2 \\ \vec{X}_1^{\,t} + \alpha \cdot \cos(\theta) \cdot \left( \left( \vec{U} - \vec{L} \right) \cdot \tau_3 + \vec{L} \right) \cdot \vec{U}_2, & Otherwise \end{cases} \tag{7}
$$

$$
= \begin{cases} \left( 1 - \frac{t}{T_{max}} \right)^{2\frac{t}{T_{max}}}, & if\ r_1 > r_2 \\ \left( \frac{t}{T_{max}} \right)^{\frac{2}{t}}, & Otherwise \end{cases} \tag{8}
$$

Equations (5)–(8) provide a mathematical model of the reference points (*RP*). Abdel-Basset assumed each cache has two reference points [25], then the *RP* matrix in Equation (5) has two columns. Equations (6) and (7) describe the two methods for solving the *RP*. Additionally, the process of the nutcracker is described as learning from past experiences continuously during retrieval; Equation (8) is used to modify *RP*, with the first term in $\alpha$ linearly decreasing and the second term linearly increasing.

$$
\vec{X}_i^{\,t+1} = min(\begin{cases} \vec{X}_i^{\,t}, & if\ f\left( \vec{X}_i^{\,t} \right) < f\left( \vec{RP}_{i,1}^{\,t} \right) \\ \vec{RP}_{i,1}^{\,t}, & Otherwise \end{cases}, \begin{cases} \vec{X}_i^{\,t}, & if\ f\left( \vec{X}_i^{\,t} \right) < f\left( \vec{RP}_{i,2}^{\,t} \right) \\ \vec{RP}_{i,2}^{\,t}, & Otherwise \end{cases}) \tag{9}
$$

$$
X_{ij}^{t+1} = \begin{cases} X_{ij}^t, & if\ \tau_3 < \tau_4 \\ X_{ij}^t + r_1 \cdot \left( X_{bestj}^t - X_{ij}^t \right) + r_2 \cdot \left( \vec{RP}_{i,1}^{\,t} - X_{ij}^t \right), & Otherwise \end{cases} \tag{10}
$$

$$
X_{ij}^{t+1} = \begin{cases} X_{ij}^t, & if\ \tau_5 < \tau_6 \\ X_{ij}^t + r_1 \cdot \left( X_{bestj}^t - X_{ij}^t \right) + r_2 \cdot \left( \vec{RP}_{i,2}^{\,t} - X_{ij}^t \right), & Otherwise \end{cases} \tag{11}
$$

$$
\vec{X}_i^{\,t+1} = \begin{cases} Equation\ (10), & if\ \tau_7 < \tau_8 \\ Equation\ (11), & Otherwise \end{cases} \tag{12}
$$

$$
\vec{X}_i^{\,t+1} = \begin{cases} Equation\ (6), & if\ \varphi > P_{a_2} \\ Equation\ (9), & Otherwise \end{cases} \tag{13}
$$

During the second exploration process of the nutcracker, Formula (9) describes the process of searching for seeds and recovering a cache in the storage area. The smaller result of the two reference points is taken as the new location, as described in Formulas (10) and (11). Based on the optimal solution $X_{bestj}^t$ and $\vec{RP}_{i,1}^{\,t}$ and $\vec{RP}_{i,2}^{\,t}$, the second development phase

is described in terms of developing corresponding areas, and using $\tau_7$, $\tau_8$ random numbers to synthesize the two reference points, as shown in Formula (12). Finally, similar to Formula (4) in the foraging and storage strategy, $P_{a_2}$ is used to balance the exploration and development process under this strategy (note that $P_{a_2}$ should be set to 0.2 based on experimental verification).

### 3.4. Implementation of NOA

As shown in the pseudo code of NOA in Algorithm 1, the implementation of NOA in the optimization process requires the initialization of the population in accordance with the set boundaries:

$$\overrightarrow{X}_{i,j}^{t} = \left( \overrightarrow{U}_j - \overrightarrow{L}_j \right) \cdot \overrightarrow{RM} + \overrightarrow{L}_j, i = 1, 2, \ldots, N, j = 1, 2, \ldots, D \tag{14}$$

---

**Algorithm 1**: Pseudo-code of NOA.

---

**Input**: population size N, the lower limits of variables $\overrightarrow{L}$, the upper limits of variables $\overrightarrow{U}$ the current number of iteration t = 0, and the maximum number of iterations $T_{max}$.
**Output**: the best solution found.
1. Initialize N nutcracker/solution using Equation (14);
2. Evaluate each solution and find the one with the best fitness in the population
3. $t = 1$; //the current function evaluation//
4. **while** ($t < T_{max}$)
5.    Generate random numbers $\sigma$ and $\sigma_1$, between 0 and 1.
6.   **If** $\sigma < \sigma_1$ //* **Foraging and storage strategy**\*//
7.     $\varphi$ is a random number between 0 and 1.
8.    **for** i = 1: N
9.     **for** j = 1: d
10.       **if** $\varphi > P_{a_1}$ /\***Exploration phase1**\*/
11.         Updating $\overrightarrow{X}_i^{t+1}$ using Equation (1)
12.      **else** /\***Exploitation phase1**\*/
13.         Updating $\overrightarrow{X}_i^{t+1}$ using Equation (3)
14.      **end if**
15.     **end for**
16.     Update the current iteration $t$ by $t = t + 1$
17.    **end for**
18.   **else** //* **Cache-search and recovery strategy** \*//
19.     Generate RP matrix using Equations (5)–(7).
20.     Generate a random number $\phi$ between 0 and 1.
21.    **for** i = 1: N
22.      **if** $\phi > P_{a_2}$ /\***Exploration phase2**\*/
23.        Updating $\overrightarrow{X}_i^{t+1}$ using Equation (9).
24.      **else** /\***Exploitation phase2**\*/
25.        Updating $\overrightarrow{X}_i^{t+1}$ using Equation (12).
26.      **end if**
27.      $t = t + 1$
28.    **end for**
29. **end while**

---

Following this, according to the foraging and storage strategies, the nutcracker population performs exploration and exploitation in first stage, and the cache search and retrieval strategies in the second stage. These two strategies are implemented in parallel and have equal probabilities. Every nutcracker represents a solution of the problem. Possible food sources in the first stage and caches in the second stage represent candidate solutions. The fitness function values to be optimized are calculated by selecting new solutions according to the strategies, and the iteration process continues to select better solutions.

During the initialization process, in addition to the population size, search boundaries, and maximum number of iterations, the NOA also requires presetting two parameters, Alpha: the percentage of attempts to avoid local optima, and $P_{rb}$: the percentage of explorations of other regions in the search space. In this paper, Alpha = 5% and $P_{rb}$ = 0.2, chosen in accordance with the recommended values in the original code, to balance the exploration and exploitation processes. In both strategies, exploration and exploitation in the solution space are equally important in regards to following the information-sharing mechanism and mutually coordinating to balance and avoid local optima.

## 4. Experimental Analysis

During the algorithm iteration, the number of iterations in each run is critical to the algorithm's performance. Too few iterations can limit the accuracy and reliability of the optimization results, and the algorithm may get stuck in local optima and fail to find the global optimal solution. In this experiment, the algorithm was set to iterate 50,000 times, and the NOA, PSO, FWA, and WOA algorithms were independently executed 30 times each, taking Aleo Solar S79Y300 monocrystalline silicon solar panels under 1000 W/m$^2$ solar radiation and 25 °C temperature as an example. The RMSE average, MD, and STD were calculated to evaluate the accuracy and reliability of the algorithm, as shown in Table 5. The best values of SDM, DDM, and TDM models for each algorithm corresponding to the best RMSE value are detailed in Tables 6–8.

**Table 5.** Test results for SDM, DDM, and TDM models.

|  |  | NOA | PSO | FWA | WOA |
|---|---|---|---|---|---|
| SDM | AVG-RMSE | 0.018868 | 2.298162 | 1.679379 | 3.061984 |
|  | MD | 0.000099 | 1.698016 | 1.698207 | 1.698016 |
|  | STD | 0.079632 | 1.658716 | 0.103535 | 2.336065 |
| DDM | AVG-RMSE | 0.488002 | 1.447042 | 1.267698 | 1.557119 |
|  | MD | 0.024675 | 1.309913 | 1.310957 | 1.672228 |
|  | STD | 0.636135 | 0.244036 | 0.131228 | 0.236466 |
| TDM | AVG-RMSE | 0.003819 | 0.250803 | 0.378707 | 0.304575 |
|  | MD | 0.003069 | 0.017604 | 0.405576 | 0.376345 |
|  | STD | 0.003877 | 0.457756 | 0.088091 | 0.338801 |
|  | Rank | 1 | 3 | 2 | 4 |

**Table 6.** Test results for SDM.

| Algorithm | $R_s$ | $R_{sh}$ | $I_{ph}$ | $I_{SD1}$ | $n_1$ | RMSE |
|---|---|---|---|---|---|---|
| NOA | 0.006562 | 5.220376 | 7.865330 | $2.9458 \times 10^{-10}$ | 0.981397 | $7.92586 \times 10^{-5}$ |
| PSO | 0.000000 | 0.116629 | 9.426077 | 0.000000 | 0.823792 | 1.698016 |
| FWA | 0.000000 | 0.153262 | 10.000000 | $3.1557 \times 10^{-10}$ | 0.999684 | 1.131198 |
| WOA | 0.000000 | 0.116653 | 9.425459 | 0.000000 | 0.960865 | 1.698016 |

**Table 7.** Test results for DDM.

| Algorithm | $R_s$ | $R_{sh}$ | $I_{ph}$ | $I_{SD1}$ | $I_{SD2}$ | $n_1$ | $n_2$ | RMSE |
|---|---|---|---|---|---|---|---|---|
| NOA | 0.006562 | 5.229866 | 7.865346 | $2.9392 \times 10^{-10}$ | $5.4180 \times 10^{-7}$ | 0.981306 | 3.641440 | $6.02460 \times 10^{-5}$ |
| PSO | 0.046217 | 3000.00000 | 8.037722 | 0.000000 | $1.0000 \times 10^{-6}$ | 1.000000 | 2.000000 | 1.309913 |
| FWA | 0.000335 | 535.789322 | 7.249937 | $4.9392 \times 10^{-10}$ | $1.1080 \times 10^{-8}$ | 0.998794 | 2.000811 | 0.817332 |
| WOA | 0.046217 | 2332.14650 | 8.037774 | 0.000000 | $1.0000 \times 10^{-6}$ | 0.112296 | 2.000000 | 1.309914 |

**Table 8.** Test results for TDM.

| Algorithm | $R_s$ | $R_{sh}$ | $I_{ph}$ | $I_{SD1}$ | $I_{SD2}$ | $I_{SD3}$ | $n_1$ | $n_2$ | $n_3$ | RMSE |
|---|---|---|---|---|---|---|---|---|---|---|
| NOA | 0.006564 | 5.210821 | 7.865464 | $2.9270 \times 10^{-10}$ | $8.2069 \times 10^{-7}$ | $9.1951 \times 10^{-7}$ | 0.981139 | 8.745928 | 8.109913 | $6.23616 \times 10^{-5}$ |
| PSO | 0.006739 | 100.00000 | 7.824180 | 0.000000 | $1.0000 \times 10^{-6}$ | $1.4708 \times 10^{-11}$ | 1.000000 | 1.691382 | 0.877124 | 0.013344 |
| FWA | 0.004407 | 62.997622 | 7.939448 | 0.000000 | $1.0000 \times 10^{-6}$ | $1.0000 \times 10^{-6}$ | 0.556490 | 2.035365 | 1.483360 | 0.106027 |
| WOA | 0.006551 | 93.154950 | 7.813773 | $4.6284 \times 10^{-10}$ | $9.3795 \times 10^{-7}$ | $5.1560 \times 10^{-9}$ | 1.000000 | 9.460711 | 9.315814 | 0.019484 |

When applying MA to optimize parameters, setting search boundaries requires a comprehensive consideration of the value range, changing trend of the objective function, feasible constraints, and the convergence and stability of the algorithm. In this paper, four algorithms were tested and analyzed, and the parameter configurations were determined based on the test results analysis. The parameter configurations are shown in Table 9.

**Table 9.** Variable Range Setting for SDM, DDM, TDM.

| Parameters | Lower Bound | Upper Bound |
|---|---|---|
| $R_s(\Omega)$ | 0 | 10 |
| $R_{sh}(\Omega)$ | 0 | 3000 |
| $I_{ph}(A)$ | 0 | 10 |
| $I_{SD1}, I_{SD2}, I_{SD3}(\mu A)$ | 0 | 1 |
| $n_1, n_2, n_3$ | 0 | 10 |

The iterative convergence graphs of SDM, DDM, and TDM algorithms are illustrated in Figure 2. The horizontal axis represents the number of iterations, and the vertical axis shows the best fitness values (on a logarithmic scale). Table 10 records the average time taken by each algorithm. A clear observation from Figure 2 is that NOA exhibits the best accuracy. Taking RMSE as an example, data from Tables 5–7 demonstrate that NOA achieves the best accuracy among the three models. Moreover, Figure 3 shows an excellent fit between the curve results obtained by the NOA and the measured data.

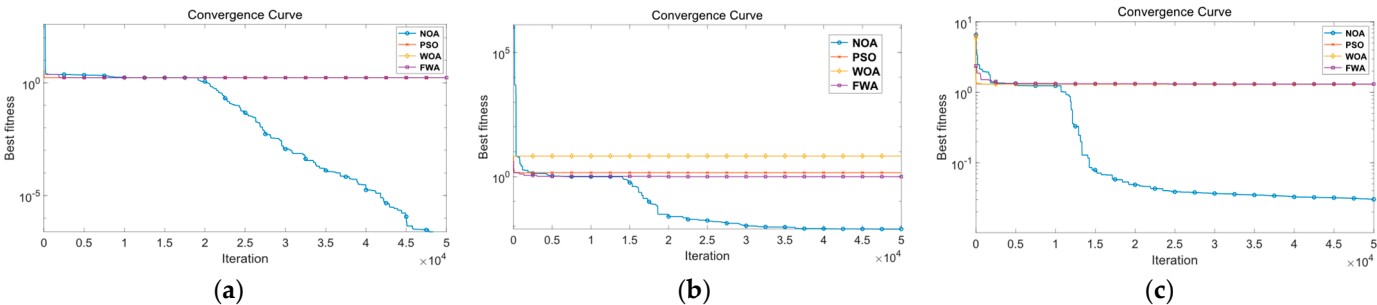

**(a)**　　　　　　　　　　　**(b)**　　　　　　　　　　　**(c)**

**Figure 2.** Convergence graph for (**a**) SDM, (**b**) DDM, (**c**) TDM.

**Table 10.** Convergence time (in seconds).

| Algorithm | SDM | DDM | TDM | AVG-Time | RANK |
|---|---|---|---|---|---|
| NOA | 406.19 | 404.87 | 729.26 | 513.44 | 1 |
| PSO | 4279.71 | 6378.25 | 4177.62 | 4945.19 | 3 |
| FWA | 4957.14 | 9037.98 | 12516.82 | 8837.32 | 4 |
| WOA | 2262.39 | 2263.33 | 4610.02 | 3045.24 | 2 |

Table 11 provides a comparative analysis of the NOA against PSO, FWA, and WOA, utilizing the nonparametric Wilcoxon rank test for independent samples [33]. The test examined the best fitness values derived from 30 independent experiments per function. The *p*-values and h-values recorded in the table offer a quantitative and qualitative understanding of NOA's performance relative to the other algorithms. The *p*-values in Table 11, all of which are below the set significance level of 0.05, suggest a significant difference in

NOA's performance and probability distribution from its counterparts. Meanwhile, the h-value indicates whether the null hypothesis is accepted (h = 1) or rejected (h = 0). Any non-rejection of the null hypothesis would imply significant differences between NOA and a certain algorithm for a specific function, as per the Wilcoxon test.

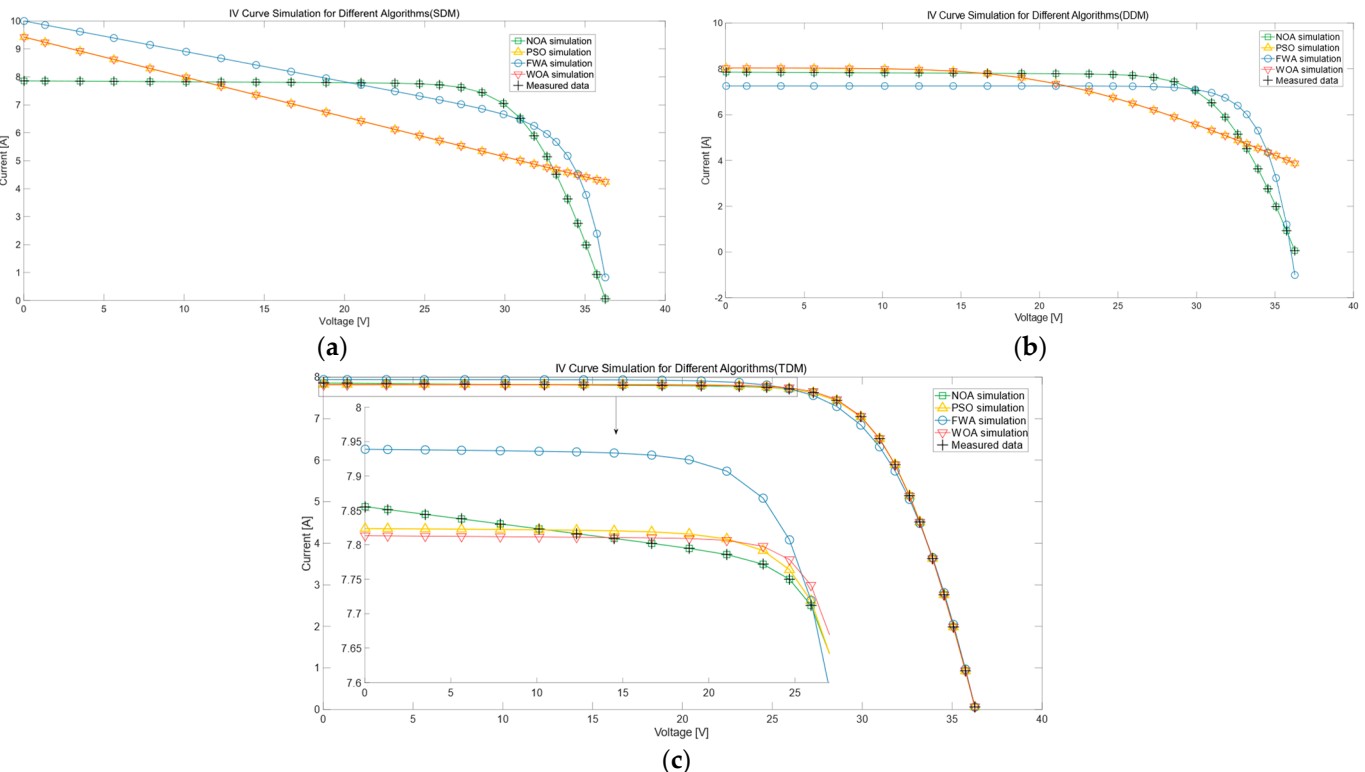

**(a)**

**(b)**

**(c)**

**Figure 3.** Simulated I–V curves of (**a**) SDM, (**b**) DDM, (**c**) TDM.

**Table 11.** Wilcoxon rank test, comparative results.

| NOA vs. | PSO | h | FWA | h | WOA | h |
|---|---|---|---|---|---|---|
| SDM | $3.01985 \times 10^{-11}$ | 1 | $1.91209 \times 10^{-9}$ | 1 | $3.01985 \times 10^{-11}$ | 1 |
| DDM | $2.97473 \times 10^{-11}$ | 1 | $1.90126 \times 10^{-9}$ | 1 | $3.00270 \times 10^{-11}$ | 1 |
| TDM | $2.50779 \times 10^{-11}$ | 1 | $2.80032 \times 10^{-11}$ | 1 | $3.00098 \times 10^{-11}$ | 1 |

Through comparative analysis, it is evident that, although PSO, WOA, and FWA demonstrate remarkable performance in various optimization problems, their effectiveness is limited in this study. Upon reaching 50,000 iterations, the results of these algorithms still fail to achieve the desired optimal solution. This phenomenon can be primarily attributed to two aspects. First, the convergence behavior of these algorithms makes it difficult for them to escape local optima in a short time, ultimately obstructing their identification of global optimal values. Second, the insufficient compatibility between these algorithms and the unique characteristics of the Aleo Solar S79Y300 monocrystalline silicon solar cell, module parameter extraction problem leads to reduced efficiency in solving this specific issue. This implies that the problem features might not align well with the fundamental assumptions or heuristic methods employed by PSO, FWA, and WOA, resulting in suboptimal performance.

By contrast, the NOA consistently outperforms popular algorithms (such as PSO, FWA, and WOA) in terms of accuracy and computational efficiency. The primary reason behind NOA's excellence is its dual search strategy approach, which expertly balances exploration and exploitation throughout the search process. As described in Section 3, the dual-strategy approach consists of two overall strategies: a foraging and storage strategy, and a caching and restoration strategy. These strategies work synergistically to facilitate a dynamic and

adaptable search process that bypass local optima and quickly converge on the expected high-quality solution. The algorithm uses random numbers in each stage of various strategies during the optimization process, enhancing its adaptability and responsiveness to the search environment, which enables it to deal with complex multi-dimensional nonlinear optimization problems effectively. Simultaneously, to address the unique challenges in the search process, both strategies include specialized exploration and development algorithms that are balanced and coordinated with each other. The exploration operator emphasizes expanding the search scope by guiding the algorithm to unknown regions of the solution space, while the development operator focuses on refining and enhancing the current best solution. The parallel integration of these strategies, along with their respective exploration and exploitation operators, fosters a highly adaptive and dynamic search process. This adaptability enables the algorithm to identify promising solution regions efficiently, and quickly converge to the global optimum, even in complex optimization scenarios, such as the TDM involving three exponential formulas and nine-dimensional variables.

### 5. Conclusions

In this paper, the applicability and accuracy of NOA for solar cell parameter estimation problem were explored, and three popular algorithms, PSO, FWA, and WOA, were applied in SDM, DDM, TDM models, respectively. Based on experimental results and comparisons, the following conclusions can be drawn: NOA is suitable for solving solar cell problems, and the results show that NOA has the best performance and the smallest error in optimizing solar cell parameters compared with the other three popular algorithms. Additionally, the average running time results show that the running time of NOA is also the shortest among the four algorithms. In this work, the best RMSE value of SDM is $7.92587 \times 10^{-5}$, that of DDM is $6.02460 \times 10^{-5}$, and that of TDM is $6.23617 \times 10^{-5}$, as shown through NOA calculation and simulation. However, according to the NFL theorem [34], future work will seek new test sets, such as arsenic gallium, perovskite material solar cells, or experimental measurements of photovoltaic modules using different technologies such as polycrystalline and thin films under different irradiation intensities and temperatures to the performance of NOA, a potential meta-heuristic algorithm.

**Author Contributions:** Writing—original draft preparation, Z.D.; writing—review and editing, H.Y., Q.Z. and L.T. All authors have read and agreed to the published version of the manuscript.

**Funding:** This research received no external funding.

**Institutional Review Board Statement:** Not applicable.

**Informed Consent Statement:** Not applicable.

**Data Availability Statement:** Not applicable.

**Conflicts of Interest:** The authors declare no conflict of interest.

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
