# Peer review of "Parameter Extraction of Solar Photovoltaic Model Based on Nutcracker Optimization Algorithm"

_applsci, doi:10.3390/app13116710_

Round 1
Reviewer 1 Report
This study uses the nutcracker optimizer algorithm for the parameter extraction of the PV model, such as single diode model, double diode model, and triple diode model of PV components. The following comments are suggested.
1. The findings of this study should be mentioned in Abstract.
2. The importance of this study is unclear; moreover, the contribution of this study should be summarized at the end of the introduction.
3. This study lacks a deep literature review, and it is suggested to review the metaheuristic algorithm introduced to solve the parameter extraction of the PV model problem. Moreover, related studies such as Quantum-based avian navigation optimizer algorithm and starling murmuration optimizer can be mentioned.
4. It is suggested to describe parameters used in equations 1-13 should be stated. The nomenclature table is recommended.
5. It is suggested to analyze how the parameters of the nutcracker optimizer algorithm are tuned to solve this problem. It is also suggested to mention which parameter tuning method is used.
6. It is suggested to compare the nutcracker optimizer algorithm with the QANA algorithm.
7. Statistical analyses such as Friedman and Wilcoxon are recommended.
8. The visualization and the quality of curves should boost.
9. The flowchart or pseudocode is recommended.
Reviewer 2 Report
The article deals with current issues. The text is relevant. I recommend the article for publication after a minor revision (see my formal comments):
Comments:
- The symbols of the physical quantities should be written in italica font, units in standard font. This graphical standard is not strictly used. See for example tables 1, 5, 8, lines 37, 83, 84, 86, …
- Lines 19, 87, 253 tables 5, 6, 7 – It is better to write number formate 7.92587·10-5, not 7.92587e-05 …
- Figure 2 – The figure caption should be on the same page like the figure.
- I don't understand Figure 3, please clarify. There are three curves, but there are 5 types of marks in the label. Some of the I-U characteristics look strange. The I-U characteristics depend on temperature according the physical theory of solids. I think, the article doi: 10.1109/JPHOTOV.2021.3108484 could be mentioned in references.
- Table 5 should not be split into two different pages.
Reviewer 3 Report
The study considers a relevant optimization problem, is well organized, and is well presented. The results obtained are quite significant. Therefore, it is a research that could be published, provided that some deficiencies that are discussed below are corrected.
More specific aspects
- The contrast of the solar panel is made assuming one thousand watts per square meter of solar irradiance at a temperature of 25 degrees Celsius. This is probably a reasonable choice, since these are frequent conditions in the geographical places of the world where more is invested in this type of energy. In any case, it should be clear why, in particular, these values have been chosen.
- The results of the experiment confirm the superiority of the selected algorithm in two aspects: a) adjustment speed, and, b) efficiency in terms of the quality of the adjustment obtained. However, it should be explained more clearly why these improvements are relevant, or if the existing algorithms were already good enough.
- The authors indicate that future research should verify this exercise with other types of photovoltaic cells. This is correct, but other types of conditions related to irradiance and the temperature under which the panels operate should also be checked.
Minor issues
- The English in general is adequate, but in some points, it requires considerable revision.
- The references are correct and current, but somewhat scarce.
correct, but needs some revision at some points.
